# HSRL: Hierarchical Spatial Reasoning with Large Language Model

## Abstract

Large language models (LLMs) have shown remarkable proficiency in general language understanding and reasoning. However, they consistently underperform in spatial reasoning, a crucial cognitive skill that severely limits their application, particularly in embodied intelligence. Inspired by the success of hierarchical learning in reinforcement learning, this paper introduces a novel method for hierarchical task decomposition in LLM spatial reasoning. Our approach leverages LLMs to break down complex spatial tasks at both the state and environment levels into more manageable sub-tasks. Specifically, we guide the LLM to identify a few key intermediate states, which are then used to generate simplified sub-environments between these key intermediate states. However, we observed that due to the LLM's lack of pre-training for spatial reasoning, it struggles to make optimal decisions during this decomposition process. To address this limitation and enhance its planning capability, we propose a novel algorithm: MCTS-Guided Group Relative Policy Optimization (M-GRPO). This algorithm integrates an MCTS-inspired exploration process and a modified, more fine-grained advantage function, enabling the model to learn optimal path planning. Experimental results demonstrate that our method substantially improves LLM performance on spatial tasks, including navigation, planning, and strategic games, achieving state-of-the-art results. This work paves the way for LLMs in real-world applications.

## 1 Introduction

Large Language Models (LLMs) have revolutionized the landscape of artificial intelligence, achieving remarkable breakthroughs across various domains, including natural language processing and scientific reasoning(Zhao et al., 2023). However, as LLMs transition into the era of embodied AI, a critical and persistent bottleneck has emerged: their inherent limitations in spatial reasoning. While Large Language Models (LLMs) excel at manipulating abstract concepts and language, they often struggle with understanding complex spatial relationships, performing efficient path planning, and engaging in sequential action reasoning(Ma et al., 2025; Chen et al., 2024). This severely limits their development and practical deployment in embodied systems.

Existing research has explored several avenues to address this challenge, yet each faces significant limitations. Prompt engineering methods like CoT(Wei et al., 2022b), ToT(Yao et al., 2023b) and ProgPrompt(Singh et al., 2023) aim to elicit reasoning through specialized prompts, but their effectiveness is capped by the model's often flawed intrinsic spatial capabilities. Fine-tuning approaches (Dao & Vu, 2025; Deng et al., 2025; Aghzal et al., 2024b) show promise but typically demand vast, expensive task-specific datasets and suffer from poor generalization to novel environments. Task decomposition strategies like HyperTree (Gui et al., 2025) and Plan-and-Act (Erdogan et al., 2025b) are primarily designed for tasks with clear, language-based "logical breaks", rendering them ill-suited for spatial reasoning problems like pathfinding that lack such linguistic segmentation. Finally, offloading planning to external, non-differentiable tools breaks the end-to-end optimization paradigm, as these tools cannot be jointly trained with the LLM's representation layer and may not be universally deployable at test time.

To overcome these limitations, we introduce Hierarchical Spatial Reasoning with LLM (HSRL), a novel hierarchical spatial reasoning paradigm inspired by Hierarchical Reinforcement Learning. The core innovation of HSRL lies in its state- and environment-based hierarchical mechanism, which is

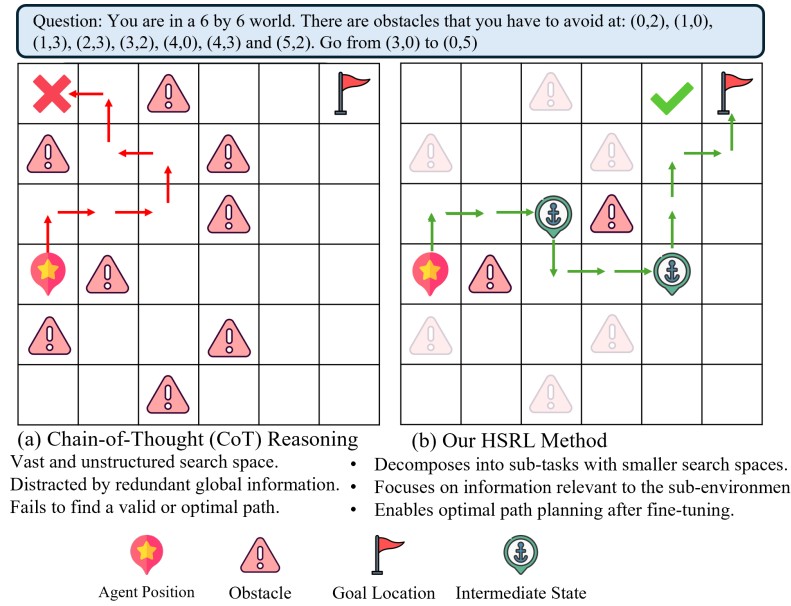

Figure 1: Comparison between CoT Reasoning and our HSRL method. **(a)** Standard CoT reasoning fails on complex spatial planning by inefficiently exploring a vast search space amidst distracting information. **(b)** In contrast, our HSRL framework succeeds by decomposing the task via key intermediate states and constructing focused sub-environments, which enables efficient and optimal planning.

fundamentally different from prior language-based decomposition methods. The framework employs a two-level hierarchy. A high-level LLM planner sets a sequence of key intermediate states (sub-goals) to break down the task. Then, for each state-to-state transition, a low-level LLM actor assumes two key roles: it first acts as an environment processor to construct a simplified, localized sub-environment, and then as a low-level LLM to execute the precise steps needed to reach the next sub-goal within that context. While this hierarchical structure provides a powerful framework for decomposition, the quality of the generated sub-goals is entirely dependent on the pre-trained high-level LLM, which often lacks the fine-grained spatial awareness needed for optimal planning. To address this, we propose an innovative online fine-tuning framework, M-GRPO, designed to enhance the high-level planner. Our approach improves planning optimality by tackling two fundamental challenges: effective exploration of the solution space and precise credit assignment for training. To achieve robust exploration, we draw inspiration from Monte Carlo Tree Search (MCTS), where the high-level LLM generates multiple candidate sequences of intermediate states, building a search tree to systematically explore diverse planning strategies. For precise credit assignment, we introduce a fine-grained advantage function, a significant departure from traditional Group Relative Policy Optimization (GRPO) which evaluates whole-trajectory values without detailed supervision. Our method calculates the advantage of each intermediate state relative to its "sibling" states (i.e., those that share a common prefix state sequence). This provides a focused and accurate training signal, enabling the LLM to learn which specific sub-goals are the most effective. Our method requires only a small amount of data and can be flexibly applied to multi-level planning tasks. In summary, this work makes the following key contributions:

- **A Novel State- and Environment-Based Hierarchical Reasoning Framework:** We introduce HSRL, a framework that presents a novel state- and environment-based decomposition paradigm for LLM spatial reasoning, departing from prevalent language-based methods. This paradigm is specifically designed to address continuous spatial problems where traditional language-based decomposition is ineffective.

- **A Novel Fine-Tuning Framework for Planning Optimality:** To address the sub-optimal planning inherent in pre-trained LLMs, we develop M-GRPO, a new fine-tuning algorithm. By integrating a Monte Carlo Tree Search exploration mechanism with a fine-grained,

node-level advantage function, our method substantially improves planning optimality with high data efficiency.

- **Comprehensive Empirical Validation of Superiority:** Through extensive experiments on large-scale navigation, object planning, and strategy game benchmarks, we demonstrate that HSRL achieves state-of-the-art performance. The results validate its significant gains over existing methods and its strong generalization across diverse task modalities.

## 2 RELATED WORK

### 2.1 SPATIAL REASONING IN LARGE LANGUAGE MODELS

Many researchers have pointed out that LLMs have weaknesses in spatial reasoning or spatial planning(Aghzal et al., 2024a;b). To address these issues, some methods leverage in-context examples and prompting techniques, such as Chain-of-Thought (CoT)(Wei et al., 2022a) and Tree-of-Thought (ToT)(Yao et al., 2023a), which have demonstrated remarkable reasoning abilities in various tasks. However, for spatial reasoning tasks, in-context learning often fails because LLMs lack spatial reasoning knowledge or their knowledge even conflicts with it.

To overcome this challenge, some studies utilize LLMs for general-purpose reasoning, converting spatial information into logical forms(Yang et al., 2023) or using them as a general pattern machine for sequence transformation(Mirchandani et al., 2023; Gong et al., 2024). Recently, other works have evaluated LLMs as a cognitive capability in navigation and planning tasks(Momennejad et al., 2023). However, these methods perform poorly in tasks requiring continuous action reasoning.

Another mainstream approach introduces closed-loop feedback mechanisms. Some works, like (Renze & Guven, 2024), use self-reflection for self-evaluation and replanning, while others adopt external feedback for reflection (Kumar et al., 2024). Furthermore, the Vision-of-Thought (VoT) method (Wu et al., 2024) materializes intermediate states to assist with reasoning. Nevertheless, this iterative feedback loop often results in high costs and inefficiency in querying or interactions.

### 2.2 HIERARCHICAL METHOD

Hierarchical reasoning breaks down decision-making tasks into multiple levels, from high-level strategic planning to low-level specific control. This decomposition reduces computational complexity by solving several less difficult sub-tasks, thus enabling the handling of tasks more challenging than direct complex reasoning. Hierarchical reasoning has achieved notable results in many reinforcement learning tasks, especially in embodied AI scenarios. For example, (Duan et al., 2020) has applied hierarchical methods to autonomous driving, allowing for smooth and safe decision-making on highways. (Lu et al., 2023) and (Zhu & Hayashibe, 2023) separate decision-making tasks into different layers, such as global path planning and local motion control. These models benefit from breaking down the decision-making process into simpler, more tractable components, enabling each layer to focus on a specific task. This enhances computational efficiency and decision accuracy in complex environments.

In recent years, hierarchical reasoning methods have also been successfully introduced into the planning tasks of LLMs. For instance, DeAR(Xue et al., 2024) imitates the human reasoning cycle by using a tree-based question decomposition approach to organize the reasoning process and break down problems into simpler sub-questions. HyperTree Planning(Gui et al., 2025) is a new paradigm that enhances LLM reasoning with a hypertree structure. It effectively breaks down intricate reasoning steps using a flexible divide-and-conquer strategy to handle diverse constraints and manage multiple distinct sub-tasks, demonstrating superior performance in complex tasks like travel planning. Plan-and-Act(Erdogan et al., 2025a) explicitly separates high-level planning from low-level execution. This framework includes a PLANNER model for generating structured high-level plans and an EXECUTOR model for translating these plans into environment-specific actions, thereby improving performance on complex multi-step tasks such as web navigation.

However, these methods only consider high-level, coarse-grained task planning and do not fully leverage the potential of hierarchical reasoning for low-level tasks that require fine motion control, such as robotic arm motion planning. Therefore, this study aims to fill this gap by solving the complex action planning problem.

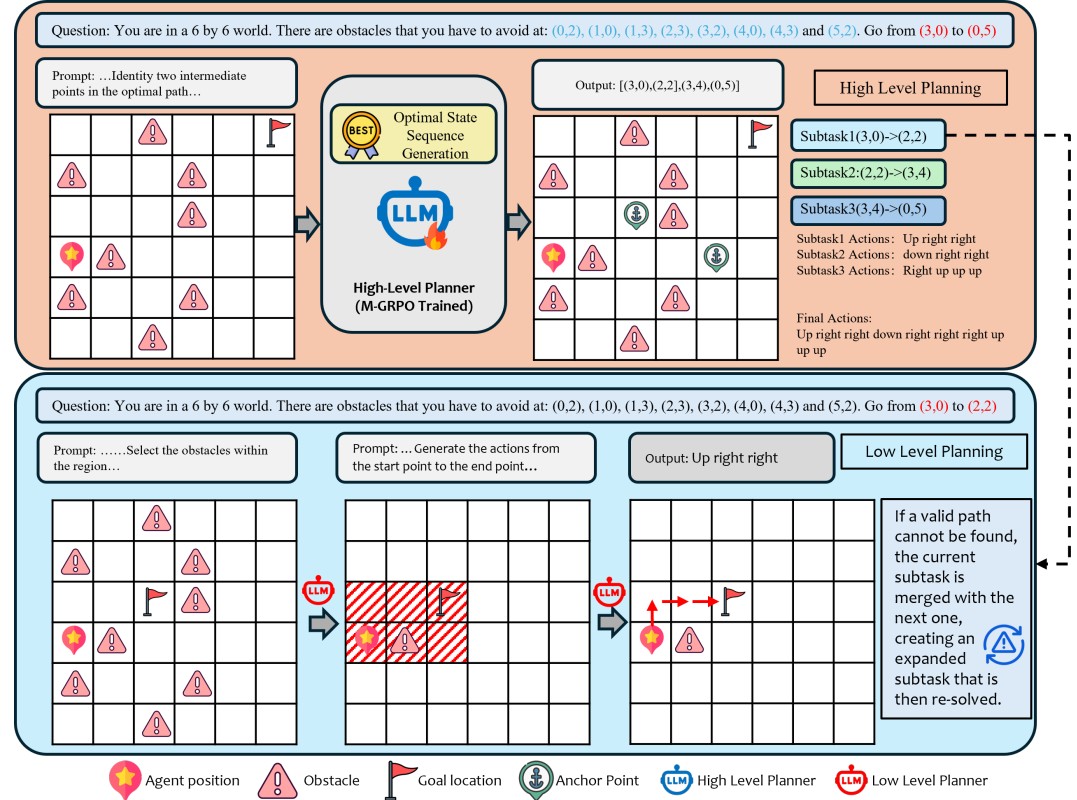

Figure 2: An overview of the HSRL framework. This framework employs a two-level hierarchical strategy. An M-GRPO trained high-level planner first identifies key intermediate states, decomposing the task into a series of sub-tasks. A low-level planner selects relevant information for the sub-task and then generates action sequences for each sub-task within a localized sub-environment. If a sub-task is unsolvable, it is merged with the next one (e.g., from the start of Subtask 1 to the end of Subtask 2) and replanned.

## 3 METHOD

In this work, we introduce the HSRL framework, as illustrated in Figure 2, to address the limitations of existing LLM-based planning methods. Our approach consists of two key components: a novel two-level hierarchical framework that decomposes complex tasks into a series of manageable sub-problems, and an innovative MCTS-guided finetuning method designed to enhance the optimality of the generated plans.

### 3.1 HIERARCHICAL PLANNING WITH STATE AND ENVIRONMENT DECOMPOSITION

Our framework leverages a two-level hierarchical decomposition strategy to break down complex planning tasks. This decomposition is applied at both the state level and the environmental level, effectively managing the complexity of the problem space.

**State-Level Decomposition via LLM.** Prior research in LLM-based path planning has shown promising results by manually decomposing tasks into sub-goals (Aghzal et al., 2024b). We extend this concept by enabling the LLM to autonomously generate these key intermediate states. Given a task's initial and final states, our method prompts the LLM to reason and generate a concise sequence of critical intermediate states. This process transforms a high-level goal into a series of state-to-state transitions, effectively simplifying the planning horizon for subsequent steps.

**Environmental-Level Decomposition and Dynamic Expansion.** After decomposing the task at the state level, much of the global environmental information becomes irrelevant noise for solving

a specific sub-task, which can hinder the reasoning process. Following the generation of the state sequence, we define a sub-task for each consecutive pair of intermediate states. For each sub-task, we create a corresponding regional environment by identifying information that is relevant to the current sub-problem (e.g., obstacles or landmarks within a localized area). This hierarchical representation allows the model to focus on a smaller, more manageable sub-environment, thereby improving efficiency and reducing the search space. If the model is unable to find a valid path within the localized environment, the scope of the sub-task is expanded. The end state is extended to the next intermediate state in the sequence, creating a larger sub-task that encompasses a broader area. This process is repeated until a solution is found or, in the worst case, the problem reverts to the original, full-scale task, ensuring robust and complete coverage of the problem space.

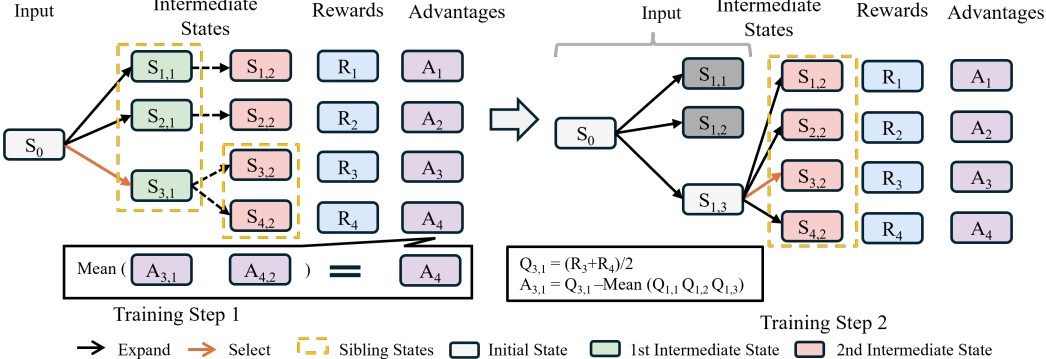

Figure 3: An overview of the M-GRPO algorithm. Starting from the initial state $S_0$, four simulation sequences are expanded in parallel. Subsequently, the advantage values of the nodes within these sequences are calculated, and a model training step is performed based on these advantages. Upon completion of the training, an optimal intermediate state is selected according to the UCT formula. This selected state then serves as the starting point for a new round of expansion and training. The entire process continues until the termination condition of MCTS is met.

### 3.2 OPTIMIZING PLANNING WITH M-GRPO

Due to an inherent lack of pre-training in spatial reasoning, LLMs often struggle to generate optimal sequences of intermediate states. However, the generation of correct intermediate states is a critical prerequisite for the success of subsequent environment decomposition and low-level action planning. To address this, we propose an online learning approach, as illustrated in Figure 3, that integrates the exploratory power of MCTS with the fine-tuning process of GRPO. This approach enables the LLM to learn and improve its planning policy during exploration.

**MCTS-Guided Exploration for Optimal State Generation** The state generation process is framed as a search problem navigated by MCTS. Within this search tree, each node represents an intermediate state, while a full sequence of nodes forms a complete trajectory, known as a completion. Starting from the initial state, the tree is built iteratively. In each iteration, the MCTS policy traverses the tree to a leaf node. From this state, the LLM is prompted to generate subsequent potential states (expansion), a reward is evaluated (simulation), and the Q-values along the path are updated (backpropagation). The selection of an intermediate state during the tree traversal is guided by the Upper Confidence bound for Trees (UCT) formula:

$$s_{\text{next}} = \underset{s' \in \text{Children}(s)}{\arg\max} \left( Q(s') + c\sqrt{\frac{\ln N(s)}{N(s')}} \right) \tag{1}$$

where $s_{\text{next}}$ is the selected next state from the set of children of the current state $s$, $Q(s')$ is the estimated value of state $s'$, $N(s)$ and $N(s')$ are the visit counts for the respective states, and $c$ is a constant controlling the level of exploration.

---

**Algorithm 1** M-GRPO Training Algorithm

---

**Require:** High-level planner $\pi_\theta$, initial state $s_0$, max iterations $N_{max}$
**Ensure:** Optimized planner $\pi_\theta$
1: $T \leftarrow$ InitializeTree($s_0$)
2: iteration $\leftarrow 0$
3: **while** iteration $< N_{max}$ **and not** IsSufficientlyDeep($T$) **do**
4:      $L \leftarrow$ SelectPromisingLeafNode($T$)
5:      $\{\tau_i\} \leftarrow$ ExpandAndSimulate($\pi_\theta, L$)                         $\triangleright$ Generate a set of new trajectories.
6:      **for** each trajectory $\tau_m$ in $\{\tau_i\}$ **do**
7:          $R_m \leftarrow$ SimulateToGoal($\tau_m$)                 $\triangleright$ Calculate the reward of each trajectory.
8:          $T \leftarrow$ Backpropagate($T, \tau_m, R_m$)        $\triangleright$ Update Q-values based on simulation results.
9:      $A_{\text{all}} \leftarrow []$                     $\triangleright$ Stores the final average advantage for each trajectory.
10:     **for** each trajectory $\tau_m$ in $\{\tau_i\}$ **do**
11:        $A_{\tau_m} \leftarrow []$                  $\triangleright$ Stores node advantages for the current trajectory.
12:        **for** each state $s$ in $\tau_m$ **do**
13:           $p \leftarrow$ GetParentNode($s$)
14:           $G_{siblings} \leftarrow$ GetChildNodes($p$)
15:           $\bar{Q} \leftarrow$ AverageQValue($G_{siblings}$)
16:           $A_s \leftarrow$ GetQValue($s$) $- \bar{Q}$          $\triangleright$ Calculate the node's fine-grained advantage.
17:           Append $A_s$ to $A_{\tau_m}$.
18:        $A(\tau_m) \leftarrow$ Average($A_{\tau_m}$)          $\triangleright$ Calculate this trajectory's average advantage.
19:        Append $A(\tau_m)$ to $A_{\text{all}}$.
20:     $\pi_\theta \leftarrow$ UpdatePolicy($\pi_\theta, A_{\text{all}}$)           $\triangleright$ Update the planner using the GRPO loss.
21:     iteration $\leftarrow$ iteration $+ 1$
22: **return** $\pi_\theta$

---

**Fine-Grained Advantage Function for Precise Policy Updates** In standard policy optimization frameworks like GRPO, the advantage function is typically computed based on the cumulative return of an entire trajectory. This coarse-grained signal poses a significant credit assignment challenge, as it fails to disambiguate the individual contributions of intermediate states. Consequently, it is difficult for the model to pinpoint which specific choices are most critical for achieving success.

To overcome this limitation, we introduce a fine-grained advantage function calculated at the intermediate state level. Our approach is tailored for a tree-search process wherein a LLM generates a set of $M$ candidate sequences (or completions), $\{\tau_1, \ldots, \tau_M\}$, for a given planning problem. Each trajectory $\tau_m$ is composed of a sequence of intermediate states, $\tau_m = (s_{m,1}, s_{m,2}, \ldots, s_{m,T_m})$.

Let $s_{m,n}$ be the $n$-th intermediate state in the $m$-th generated sequence. We estimate its corresponding state-value, or Q-value $Q_{m,n}$, as the mean empirical return from all Monte Carlo simulations that traverse this state. Specifically, if $W_{m,n}$ is the sum of cumulative rewards from all visits to state $s_{m,n}$ and $N_{m,n}$ is its total visit count, the Q-value is given by:

$$Q_{m,n} = \frac{W_{m,n}}{N_{m,n}} \tag{2}$$

We then define the advantage of a specific state, $A_{m,n}$, relative to its "sibling" states—i.e., the set of other candidate states $\{s_{j,n}\}_{j=1}^{M}$ that share a common prefix sequence. The state-level advantage is formulated as:

$$A_{m,n} = Q_{m,n} - \text{Mean}(Q_{\text{siblings}}) \tag{3}$$

where the second term represents the mean Q-value across all sibling states at depth $n$. This formulation directly quantifies how much better the choice leading to $s_{m,n}$ is compared to the average of alternative choices at that decision point. A deliberate design choice is the omission of reward normalization. As the LLM often generates identical optimal completions, forgoing normalization prevents "reward hacking," where the value of a superior path could be artificially deflated due to its high frequency of generation.

Finally, to align with the GRPO framework, we compute a single advantage value for each trajectory by averaging the advantages of all its constituent intermediate states. For a trajectory $\tau_m$ of length

$T_m$, its overall advantage $A(\tau_m)$ is calculated as:

$$A(\tau_m) = \frac{1}{T_m} \sum_{n=1}^{T_m} A_{m,n} \tag{4}$$

This trajectory-level advantage $A(\tau_m)$ is then used as the training signal within the GRPO loss function. This fine-grained approach to advantage calculation provides a more precise and informative signal, enabling the model to learn not only which overall sequences are effective, but also to discern the value of the specific intermediate states that are most critical for constructing an optimal plan. We present the full pseudo-code in Algorithm 1

## 4 EXPERIMENTS

### 4.1 EXPERIMENTAL SETUP

For our experimental setup, we designated the M-GRPO trained Qwen3-4B-Instruct-2507 as our high-level planner. The untrained version of the same model served as both the low-level planner and the environment planning model.

**Datasets** We evaluate our HSRL framework across three planning benchmarks with increasing difficulty to test its performance and generalization. First, we use Maze Navigation(Valmeekam et al., 2023), a classical task on a dataset of 1,090 $10 \times 10$ grids, partitioned into 668 training and 422 testing instances. Second, to assess out-of-distribution (OOD) generalization, we employ the Blocksworld benchmark(Saha et al., 2025), whose test set is intentionally more complex, featuring more blocks and requiring longer plans (7–10 steps) than the training set (1–6 steps). Finally, we validate our framework on the novel and highly challenging GameTraversalBenchmark (GTB)(Nasir et al., 2024). This benchmark contains 150 diverse maps with multiple objectives and paths exceeding 100 steps. As GTB lacks a training set, we evaluate our Maze-trained model in a zero-shot transfer setting to test its capabilities on complex, unseen tasks.

**Baselines** We compare HSRL against a diverse set of representative baselines. First, we compare it with foundational reasoning strategies, including the classic Chain-of-Thought (CoT)(Wei et al., 2022a) and ReAct(Yao et al., 2023c), which interleaves reasoning traces with actions for improved synergy. We also include advanced reasoning and self-reflection methods like Inner Monologue(Huang et al., 2023), which enhances internal thought processes, and Reflexion(Shinn et al., 2023), which uses iterative self-correction to refine plans. For direct planning, we use Prog-Prompt(Singh et al., 2023) as a strong representative of in-context learning-based approaches. Furthermore, we contrast HSRL with search-based methods like Tree Planner(Hu et al., 2024) and the hierarchical planner HyperTree(Gui et al.), the latter of which is known to have limitations on spatial reasoning tasks. Finally, we include System-1.x(Saha et al., 2025), a powerful baseline meticulously fine-tuned on tasks similar to ours, which employs a controller to switch between "fast-thinking" and "slow-thinking" modes.

**Evaluation metrics** We evaluate our model's planning ability using metrics tailored to each benchmark. For the classical Maze and Blocksworld tasks, we measure the Completion Rate (CR), which is the percentage of successfully solved instances, and the Optimal Rate (OR), defined as the percentage of completed tasks where the plan length matches the shortest path computed by an A* search. For the more complex GameTraversalBenchmark (GTB), we adopt its official metrics. The primary metric is the GTB Score, a composite measure that assesses performance based on goal proximity, path length, and generation errors (see Appendix B.1). Additionally, we report Top-5 Accuracy, the fraction of tasks where the agent's final position is within five tiles of the target, to evaluate approximate success in large-scale maps.

### 4.2 IMPLEMENTATION DETAIL

**M-GRPO Finetuning.** We fine-tuned the Qwen3-4B-Instruct-2507 model with the same hyperparameters as the GRPO algorithm. The configuration used the AdamW Optimizer with $\beta_1 = 0.9$ and $\beta_2 = 0.999$, and a Learning Rate of $1 \times 10^{-6}$ with Cosine decay scheduling. The Epoch Number was set to 1 and the Batch Size was 1. For the inference phase, the Temperature was set to 1.0 and the Num generations was 8.

**Reward Function.** We designed a complex reward function based on the degree of length matching with the $A^*$ path and point-wise weighted rewards to guide the training of M-GRPO, as detailed in Appendix C.1.

Table 1: Comparison of HSRL against baseline methods across various benchmarks. Performance is measured by goal achievement, optimality, and other task-specific scores. The best performance in each column is highlighted in **bold**.

| Method | Maze (size 10×10) | | Blocksworld (5-7 blocks) | | GTB | |
|---|---|---|---|---|---|---|
| | CR(%) ↑ | OR(%) ↑ | CR(%) ↑ | OR(%) ↑ | GTB Score ↑ | Top 5 Acc. (%) ↑ |
| Direct Answer | 23.69 | 23.45 | 6.50 | 6.00 | 23.61 | 25.96 |
| CoT | 43.12 | 38.39 | 10.50 | 8.00 | 26.58 | 31.61 |
| Reflexion | 45.02 | 37.91 | 15.00 | 8.50 | 29.34 | 37.76 |
| ReAct | 53.80 | 26.06 | 8.00 | 3.00 | 20.40 | 39.85 |
| ProgPrompt | 34.60 | 33.41 | 9.50 | 6.00 | 22.45 | 26.12 |
| Inner Monologue | 54.03 | 34.60 | 4.00 | 0.00 | 19.18 | 21.41 |
| System-1.x | 54.74 | 36.02 | 27.00 | 14.50 | 27.73 | 30.01 |
| HyperTree | 37.91 | 22.98 | 8.00 | 3.50 | 25.81 | 26.67 |
| Tree Planner | 39.10 | 27.01 | 7.00 | 4.00 | 25.28 | 25.46 |
| **HSRL (Ours)** | **60.43** | **46.44** | **29.50** | **18.00** | **30.65** | **40.29** |

## 4.3 RESULTS ANALYSIS

The experiments results, detailed in Table 1, clearly demonstrate the effectiveness of our hierarchical planning and search framework.

**HSRL significantly improves task performance.** Our method HSRL achieves state-of-the-art (SOTA) performance across all metrics on all tasks. For instance, in the Maze ($10 \times 10$) task, our method achieves a goal completion rate of 60.43%, markedly outperforming other methods such as System-1.x (54.74%) and ReAct (53.80%). This advantage is even more pronounced in the more complex Blocksworld task, where our method's 29.50% completion rate far exceeds all baselines. This pattern of superiority extends to the challenging GameTraversalBenchmark (GTB), where HSRL achieves the highest GTB Score of 30.65, surpassing strong competitors like Reflexion (29.34). Furthermore, it also secures the best Top 5 Accuracy at 40.29%, demonstrating its robustness in complex, large-scale planning environments.

**Superior Solution Optimality.** Beyond merely completing a task, the ability to generate optimal (or near-optimal) paths is a crucial measure of a planning model's intelligence. On the optimal rate metric for the Maze and Blocksworld tasks, our method achieves optimality rates of $46.44\%$ and $18.00\%$, respectively, the highest among all compared methods. This result indicates that by combining the forward-search capabilities of MCTS with the general knowledge of large models, our framework can explore the solution space more thoroughly, effectively avoiding local optimal to devise more efficient and concise solutions.

Table 2: Ablation study on the core components of our HSRL framework. We progressively remove key modules from our full model, HSRL (Ours), to evaluate their individual contributions across three distinct benchmarks. Best performance in each column is highlighted in **bold**.

| Model Configuration | Maze (10×10) | | Blocksworld (5-7 blocks) | | GTB | |
|---|---|---|---|---|---|---|
| | CR(%) ↑ | OR(%) ↑ | CR(%) ↑ | OR(%) ↑ | GTB Scores ↑ | Top 5 Acc.(%) ↑ |
| State-Hierarchical Only | 50.24 | 13.03 | 10.00 | 9.50 | 26.16 | 29.85 |
| HSRL (Untrained) | 54.50 | 14.22 | 12.50 | 9.50 | 26.96 | 32.38 |
| HSRL (w/o MCTS) | 55.21 | 45.97 | 28.00 | 15.00 | 27.80 | 38.09 |
| **HSRL (Ours)** | **60.43** | **46.44** | **29.50** | **18.00** | **30.65** | **40.29** |

**Cross-Task Robustness and Generalization.** The value of a general-purpose planning model lies in its cross-task generalization capability. As shown in the table, our method performs exceptionally well on the classical spatial reasoning tasks of Maze and Blocksworld, and the complex world

knowledge required for the Game Travel Benchmark. In contrast, some baselines exhibit strong task-specific biases; for example, while ReAct performs reasonably well in Maze, its completion rate plummets to just 3.00% in Blocksworld. This comparison validates the robustness of our hierarchical framework, which consistently decomposes complex problems into manageable sub-goals for effective problem-solving, irrespective of the task modality. Moreover, our excellent performance in Blocksworld demonstrates strong out-of-distribution generalization capabilities.

### 4.4 FURTHER ANALYSIS

**Ablation Study.** To validate the contribution of each component, we conducted a comprehensive ablation study (Table 2). The results reveal a clear hierarchy of importance. Removing the MCTS module (HSRL w/o MCTS) leads to a notable decline in both success and optimality, confirming that its systematic, forward-looking search is crucial for exploring diverse solution pathways and avoiding tempting local optima. Further removing the M-GRPO policy optimization (HSRL Untrained) causes a precipitous performance collapse, especially in optimality (e.g., Maze optimality plummets from 45.97% to 14.22%). This demonstrates that M-GRPO is the core engine that translates the rich search experience from MCTS into a refined planning intuition, endowing the LLM with the ability to generate high-quality, task-aligned sub-goals. Finally, the performance of the State-Hierarchical Only model also significantly surpasses direct answering methods, and the inclusion of environment-hierarchical approaches effectively improves task completion.

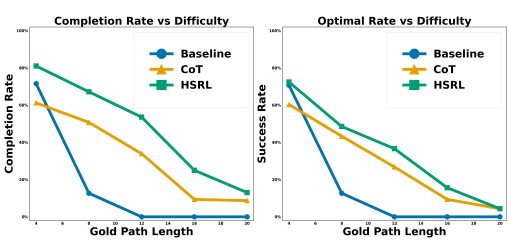

Figure 4: Performance degradation as task difficulty increases. HSRL shows greater robustness.

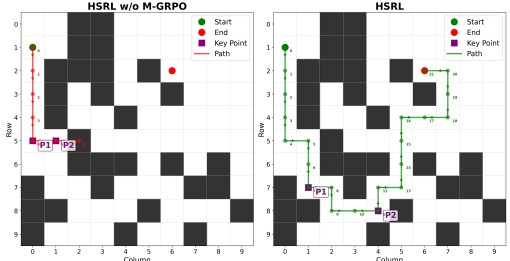

Figure 5: Qualitative comparison of a generated plan before (left) and after (right) M-GRPO training.

**Robustness and Qualitative Insights.** Further analysis highlights HSRL's robustness. As shown in Figure 4, HSRL's performance degrades far more gracefully with increasing task difficulty compared to baselines, maintaining a substantial and reliable advantage on the most challenging Maze instances. This resilience is not merely statistical; it stems from a fundamental improvement in high-level planning quality. A qualitative comparison in Figure 5 illustrates the underlying mechanism: while an untrained model generates ill-conceived sub-goals leading to a failed plan, the M-GRPO trained HSRL produces a strategic and successful path. This synergy between quantitative robustness and qualitative intelligence validates our framework's effectiveness in complex planning scenarios where long-horizon reasoning is paramount.

## 5 CONCLUSION

This study introduces HSRL, a novel hierarchical reasoning framework designed to address the deficiency of LLMs in spatial reasoning. The framework simplifies complex tasks into manageable sub-tasks through a dual decomposition of state and environment. To optimize planning capabilities, we designed the M-GRPO algorithm, which integrates the exploratory power of MCTS with a more fine-grained advantage function, significantly enhancing planning quality. Experiments demonstrate that HSRL achieves SOTA performance across multiple benchmarks, including navigation, object planning, and strategic games, substantially surpassing existing methods, particularly in task completion rates and path optimality. This work opens a new path for the application of LLMs in complex physical worlds, such as embodied intelligence.

ETHICS STATEMENT

Besides reflecting possible issues such as bias and discrimination inherited from pre-training data of large language models(Weidinger et al., 2021), our approach does not address ethical or societal concerns.

REPRODUCIBILITY STATEMENT

We include the experimental prompts in the Appendix D and provide the source code in the supplementary material to support reproducibility.

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

# A    THE USE OF LARGE LANGUAGE MODELS (LLMS)

The use of the LLM was strictly limited to polishing the language, correcting grammatical errors and typos, and assisting with formatting. All core research ideas, experimental design, analysis of results, and the final conclusions were conceived and executed solely by the authors. The authors take full responsibility for the entire content of this paper.

# B    MORE BACKGROUND

## B.1    GTB SCORE

$$\text{GTB\_Score} = \frac{1}{M} \sum_{m=1}^{M} \frac{\left( R^{(m)} - \text{LLM}_{PL}^{(m)} - \varepsilon^{(m)} \right) - R_{\min}^{(m)}}{R_{\max}^{(m)} - R_{\min}^{(m)}} \tag{5}$$

where:

- $M$ = total number of maps in the dataset.
- $R^{(m)}$ = reward obtained for map $m$, determined by the final distance $d$ to the objective:

$$R^{(m)} = \begin{cases} +200, & d = 0 \\ +100, & d = 1 \\ +50, & d \in [2,3] \\ +25, & d \in [3,5] \\ -50, & d \in [5,8] \\ -100, & d \geq 8 \end{cases}$$

- $\text{LLM}_{PL}^{(m)}$ = path length taken by the LLM agent on map $m$.
- $\varepsilon^{(m)}$ = total generation errors made by the LLM on map $m$.
- $R_{\max}^{(m)} = 200 - A_{PL}^*(m)$, the maximum achievable reward (perfect path with no errors), where $A_{PL}^*(m)$ is the optimal path length computed by an A* agent.
- $R_{\min}^{(m)} = -100 - A_{PL}^*(m) - \varepsilon_{\max}^{(m)}$, the minimum achievable reward (farthest position, maximal path cost, and maximal errors).

# C    MORE IMPLEMENTATION DETAILS

## C.1    M-GRPO REWARD

For a sampled completion $completion_i$, we parse its anchor list $A_i = [a_{i1}, a_{i2}, \ldots, a_{in}]$. If the anchor list cannot be parsed, we directly assign a fixed penalty; otherwise, the reward score is

computed by a signed power transformation to enlarge the margin between high- and low-quality completions:

$$R_i = \begin{cases} \texttt{PARSE\_FAIL\_PENALTY}, & \text{if } A_i = \emptyset, \\ \text{sign}(z_i) \, |z_i|^p, & \text{otherwise}, \end{cases} \tag{6}$$

where $\text{sign}(z_i)$ preserves the direction of $z_i$, and $p > 1$ amplifies its magnitude non-linearly.

The raw score $z_i$ aggregates the quality of anchors visited by a trajectory, while discouraging the use of overly many anchors through a penalty term:

$$z_i = \sum_{a \in A_i} \overline{r}(a) - \alpha \cdot \max\big(0, \, |A_i| - A_{\text{expected}}\big). \tag{7}$$

To evaluate each anchor consistently, we first assign each completion an initial reward $r_i$ according to its alignment with the Manhattan distance of the optimal $A^*$ path. :

$$r_i = BASIC\_QUALITY\_SCORE - \Big| \sum_{a_i \in A_i} |a_{i+1} - a_i| - \sum_{\hat{a}_i \in A^*} |\hat{a_{i+1}} - \hat{a}_i| \Big|. \tag{8}$$

Each anchor reward $\overline{r}(a)$ is then defined as the average quality of all completions that pass through it, reflecting a consensus measure across different trajectories:

$$\overline{r}(a) = \frac{\sum\limits_{i:\, a \in A_i} r_i}{|\{\, i \mid a \in A_i \,\}|}. \tag{9}$$

## D  PROMPTS AND EXAMPLES

756
757
758
759
760
761
762
763
764

---

**High-level Planning Prompt for Maze**

### Role
You are an expert high-level path planner. You must strictly adhere to the requirements outlined in the system instructions and tasks I have provided to you.

### Instructions
1. Your task is to plan a feasible, obstacle-free path for a single agent in a given 10x10 grid environment, from a start to an end point.
2. The path should be defined by a series of key anchor point coordinates.
3. You must identify exactly {{num_anchors}} feasible intermediate anchor points for the given task. These anchor points should be the key turning points of the path used to navigate around obstacles or toward the goal.

### Anchor Point Selection Strategy
- The path does not have to be the shortest path. The priority is feasibility and safety (avoiding all obstacles).
- Explore multiple valid paths and select a reasonable one to define your anchor points.
- Anchor points should be strategically located at important positions around obstacles.

### Output Format
- You must strictly follow the format below to output the list of anchor points.
- Do not provide any explanation or text other than the final trajectory list.
- Directly output the result in the given format:
<trajectory for planning> = [(start_x, start_y), (anchor_1_x, anchor_1_y), ..., (end_x, end_y)]
---
### Examples

**Example 1:**
Task: You are in a 10 by 10 world. There are obstacles that you must avoid at: (4,7), (8,6), (3,3), (9,5), (8,9), (1,1), (5,4), (1,3), (9,9), (4,1), (5,7), (1,6), (9,0), (8,3), (0,0), (7,1), (4,6), (5,0), (2,5) and (4,0). Go from (2,1) to (0,2).
<trajectory for planning> = [(2,1),(2,2),(1,2),(0,2)]

**Example 2:**
Task: You are in a 10 by 10 world. There are obstacles that you must avoid at: (0,7), (3,2), (0,4), (3,4), (4,6), (7,2), (7,3), (2,0), (3,9), (9,3), (8,2), (9,5), (8,4), (7,5), (4,8), (5,2), (5,5), (7,8), (6,3) and (9,8). Go from (6,8) to (6,1).
<trajectory for planning> = [(6,0),(6,4),(5,3),(6,1)]

**Example 3:**
Task: You are in a 10 by 10 world. There are obstacles that you must avoid at: (8,5), (7,2), (1,7), (2,0), (3,2), (5,0), (1,9), (3,3), (3,6), (4,7), (0,3), (5,7), (5,3), (4,6), (2,8), (4,3), (9,0), (7,5), (5,5) and (8,9). Go from (0,8) to (7,1).
<trajectory for planning> = [(0,8),(0,4),(1,1),(7,1)]
---

### Task to Solve
Task: You are in a 10 by 10 world. There are obstacles that you must avoid at: (9,6), (1,0), (3,7), (4,4), (9,1), (4,0), (3,4), (8,9), (7,1), (5,1), (3,6), (4,9), (4,8), (0,1), (6,4) and (0,0). Go from (4,1) to (6,8).
<trajectory for planning> = [(4,1),(5,3),(5,6),(6,8)]

---

800
801
802
803
804
805
806
807
808
809

810
811
812
813
814
815
816
817
818
819
820
821
822
823
824
825
826
827
828
829
830
831
832
833
834
835
836
837
838
839
840
841
842
843
844
845
846
847
848
849
850
851
852
853
854
855
856
857
858
859
860
861
862
863

**Environment Decomposition Prompt for Maze**

```
Given the following list of coordinate points:
[(9,6), (1,0), (3,7), (4,4), (9,1), (4,0), (3,4), (8,9), (7,1), (5,1), (3,6), (4,9), (4,8), (0,1),
(6,4), (0,0)]

Please select all points from the list that satisfy the following condition:
The x-coordinate (x) of the point must be within the closed interval from 4 to 5.

Please strictly follow this format for the output, including only the selected points:
### Output Format
<obstacles> = [ (x1, y1), (x2, y2), ..., (xn, yn)]

<assistant>
<obstacles> = [(4,4),(4,0),(5,1),(4,9),(4,8)]
```

**Environment Decomposition Prompt for Maze**

```
Given the following list of coordinate points:
[(9,6), (1,0), (3,7), (4,4), (9,1), (4,0), (3,4), (8,9), (7,1), (5,1), (3,6), (4,9), (4,8), (0,1),
(6,4), (0,0)]

Please select all points from the list that satisfy the following condition:
The y-coordinate (y) of the point must be within the closed interval from 1 to 3.

Please strictly follow this format for the output, including only the selected points:
### Output Format
<obstacles> = [ (x1, y1), (x2, y2), ..., (xn, yn)]

<assistant>
<obstacles> = [(9,1),(7,1),(5,1),(0,1)]
```

864
865
866
867
868
869
870
871
872
873
874
875
876
877

**Low-level Execution Prompt for Maze**

```
### Role
You are an low-level path planner located in a 10 by 10 world. You must strictly adhere to the requirements
of the tasks I have provided to you.

### Environment
Provide a sequence of actions to navigate a world to reach a goal. (0,0) is located in the upper-left corner
and (9,9) lies in down-right corner.

### Rules
- <left = (0,-1)>
- <right = (0,+1)>
- <up = (-1,0)>
- <down = (+1,0)>

### Output Format
Actions = [action_0 action_1 … action_n]

Here are some examples:

###
Task: You are in a 10 by 10 world. There are obstacles that you must avoid at: (2,1). Go from (0,1) to (3,4).

Actions = [right right right down down down]

###
Task: You are in a 10 by 10 world. There are obstacles that you must avoid at: (1,5) and (1,2). Go from (5,4)
to (0,5).

Actions = [up up up up up right]

###
Task: You are in a 10 by 10 world. There are obstacles that you must avoid at: (0,3), (2,5) and (5,2). Go
from (4,2) to (0,5)

Actions = [up up up right right up right]

### Task to Solve subtask_1

Task: You are in a 10 by 10 world. There are obstacles that you must avoid
at:(4,4),(4,0),(5,1),(4,9),(4,8),(9,1),(7,1),(5,1),(0,1). Go from (4,1) to (5,3).

Actions = [right right down]
```

878
879
880
881
882
883
884
885
886
887
888
889
890
891
892
893
894
895
896
897
898
899
900
901
902
903
904
905
906
907
908
909
910
911
912
913
914
915
916
917

918
919
920
921
922
923
924
925
926
927
928
929
930
931
932
933
934
935
936
937
938
939
940
941
942
943
944
945
946
947
948
949
950
951
952
953
954
955
956
957
958
959
960
961
962
963
964
965
966
967
968
969
970
971

## High-level Planning Prompt for Blocksworld

```
### Role
You are a high-level Blocksworld planner. You must strictly adhere to the requirements outlined in
the system instructions and tasks I have provided to you.

### Instructions
1. Plan a feasible sequence of block configurations from an initial state to a goal state.
2. Define the plan with exactly {{num_anchors}} intermediate stack states.
3. Intermediate states should be key subgoals (e.g., clearing a block or forming partial stacks).

### Strategy
- The sequence need not be shortest; feasibility and clarity are the priority.
- Choose anchor states that mark meaningful progress toward the goal.

### Output Format
- You must strictly follow the format below to output the list of anchor states.
- Do not provide any explanation or text other than the final output list.
- Directly output the result in the given format:
Output = [initial_state, anchor_state1, ..., goal_state]
---
### Examples

**Example 1:**
The initial state:
A is on the table. B is on A. B is clear.
The goal is:
B is on the table. A is on B. A is clear.

Output = ['A is on the table. B is on A. B is clear.','A is on the table. A is clear. B is on the
table. B is clear.','B is on the table. A is on B. A is clear.']

**Example 2:**
The initial state:
C is on the table. D is on C. D is clear.
The goal is:
C is on D. D is on the table. D is clear.

Output = ['C is on the table. D is on C. D is clear.','C is on the table. C is clear. D is on the
table. D is clear.','C is on D. D is on the table. D is clear.']

**Example 3:**
The initial state:
B is on the table. C is on B. A is on C. A is clear.
The goal is:
B is on the table. B is clear.
C is on B. A is on C. A is clear.

Output = ['B is on the table. C is on B. A is on C. A is clear.','B is on the table. B is clear. C
is on the table. C is clear. A is on the table. A is clear.','B is on the table. C is on B. C is
clear. A is on the table. A is clear.']
---
### Task to Solve
The initial state:
B is on the table. B is clear.
D is on the table. C is on D. E is on C. A is on E. A is clear.

The goal is:
C is on the table. E is on C. D is on E. B is on D. A is on B. A is clear.

Output = ['The hand is empty. B is on the table. B is clear. D is on the table. C is on the table. C
is clear. E is on the table. E is clear. A is on the table. A is clear.', 'C is on the table. E is
on C. D is on E. B is on the table. B is clear. A is on the table. A is clear.']
```

**Environment Decomposition Prompt for Blocksworld**

```
Given the 2 state of the same group of Blocks:
<1>'B is on the table. B is clear. D is on the table. C is on the table. C is clear. E is on the
table. E is clear. A is on the table. A is clear.'
<2>'C is on the table. E is on C. D is on E. B is on the table. B is clear. A is on the table. A is
clear.'

Please select all state pairs (including block-block pairs, block-table pairs, and block-clear
pairs) whose relative positions remain unchanged.

Please strictly follow this format for the output, including only the selected pairs:
### Output Format
<pairs> = [(Object1,Object2),(Object3,Object4),...]

<assistant>
<pairs> = [(B,table),(A,table),(C,table)]
```

**Low-level Execution Prompt for Blocksworld**

```
You are a Blocks World action planner. You will be given an initial state and a goal state.
Provide a sequence of actions to move the blocks to reach a goal similarly to the examples below.
Do not include any extra text or explanations.

Here are some examples:
###
The initial state:
The hand is empty.
B is on the table. A is on B. C is on A. C is clear.

The goal is:
B is on the table. A is on B. A is clear.
C is on the table. C is clear.

<Observation>: B is still on table; A is still on B
Actions: Move C from A to table

###
The initial state:
The hand is empty.
B is on the table. C is on B. D is on C. A is on D. A is clear.

The goal is:
A is on the table. A is clear.
C is on the table. B is on C. D is on B. D is clear.

<Observation>: A is still clear
Actions: Move A from D to table | Move D from C to table | Move C from B to table | Move B from
table to C | Move D from table to B

Now, here is your task:
###
The initial state:
B is on the table. B is clear.
D is on the table. D is clear.
C is on the table. C is clear.
E is on the table. E is clear.
A is on the table. A is clear.

The goal is:
C is on the table. E is on C. D is on E. D is clear.
B is on the table. B is clear.
A is on the table. A is clear.

<Observation>: B is still on table; A is still on table; C is still on table
Actions: Move E from table to C | Move D from table to E
```

1026
1027
1028
1029
1030
1031
1032
1033
1034
1035
1036
1037
1038
1039
1040
1041
1042
1043
1044
1045
1046
1047
1048
1049
1050
1051
1052
1053
1054
1055
1056
1057
1058
1059
1060
1061
1062
1063
1064
1065
1066
1067
1068
1069
1070
1071
1072
1073
1074
1075
1076
1077
1078
1079

**High-level Planning Prompt for GTB**

### Role
You are an expert high-level planner for a 2D-gird game. You must strictly adhere to the requirements outlined in the system instructions and tasks I have provided to you.

### Instructions
1. Your task is to plan a feasible sequence of moves in a {{world_height}}×{{world_length}} grid environment, from a start state to a goal state.
2. The plan should be defined by a series of key anchor coordinates.
3. You must identify exactly {{num_anchors}} feasible anchor states for the given task. These should be important turning points or subgoals used to navigate around obstacles or toward objectives.

### Anchor State Selection Strategy
- The plan does not need to be the shortest; the priority is feasibility and safety (avoiding all obstacles).
- Anchor states should be strategically located at key subgoals: clearing an obstacle, moving around blocking tiles, or forming partial progress toward objectives.
- Explore multiple valid strategies and select one reasonable plan.

### Reward Context
- Rewards are given as follows:
  {{reward_design}}
  {{reward_feedback}}
- You are also given information about your previous attempt:
  - Actions generated: {{total_actions[list(objective_tile_dict.keys())[i]]}}
  - Start position: {{prev_protagonist_position}}
  - End position: {{protagonist_position}}
  - Distance from objective: {{distance_from_objective}}
  - Objective location: {{list(objective_tile_dict.values())[i]}}
  - GTB Reward received: {{reward_this_objective[list(objective_tile_dict.keys())[i]]}}

### Output Format
- Strictly output the result in the following format, without any explanation:
<trajectory for planning> = [(start_x, start_y), (anchor_1_x, anchor_1_y), ..., (end_x, end_y)]
---
### Examples

**Example 1:**
Task: You are in a 10 by 10 world. There are obstacles that you have to avoid at: (4,7), (8,6), (3,3), (9,5), (8,9), (1,1), (5,4), (1,3), (9,9), (4,1), (5,7), (1,6), (9,0), (8,3), (0,0), (7,1), (4,6), (5,0), (2,5) and (4,0). Go from (2,1) to (0,2).
<trajectory for planning> = [(2,1),(2,2),(0,2)]

**Example 2:**
Task: You are in a 10 by 10 world. There are obstacles that you have to avoid at: (0,7), (3,2), (0,4), (3,4), (4,6), (7,2), (7,3), (2,0), (3,9), (9,3), (8,2), (9,5), (8,4), (7,5), (4,8), (5,2), (5,5), (7,8), (6,3) and (9,8). Go from (6,8) to (6,1).
<trajectory for planning> = [(6,0),(6,4),(5,3),(4,2),(6,1)]

**Example 3:**
Task: You are in a 10 by 10 world. There are obstacles that you have to avoid at: (8,5), (7,2), (1,7), (2,0), (3,2), (5,0), (1,9), (3,3), (3,6), (4,7), (0,3), (5,7), (5,3), (4,6), (2,8), (4,3), (9,0), (7,5), (5,5) and (8,9). Go from (0,8) to (7,1).
<trajectory for planning> = [(0,8),(0,4),(1,1),(7,1)]
---
### Task to Solve
Task: {{task}}
<trajectory for planning> =

**Environment Decomposition Prompt for GTB**

```
Given the following list of coordinate points:
[{obstacles_this_object}]

Please select all points from the list that satisfy the following condition:
The x-coordinate (x) of the point must be within the closed interval from {x_this_object_min} to
{x_this_object_max}.

Please strictly follow this format for the output, including only the selected points:
### Output Format
<obstacles> = [ (x1, y1), (x2, y2), ..., (xn, yn)]

<assistant>
<obstacles> =
```

**Environment Decomposition Prompt for GTB**

```
Given the following list of coordinate points:
[{obstacles_this_object}]

Please select all points from the list that satisfy the following condition:
The y-coordinate (y) of the point must be within the closed interval from {y_this_object_min} to
{y_this_object_max}.

Please strictly follow this format for the output, including only the selected points:
<obstacles> = [ (x1, y1), (x2, y2), ..., (xn, yn)]

<assistant>
<obstacles> =
```

**Low-level Execution Prompt for GTB**

```
### Role
You are an low-level path planner located in a {world_height} by {world_width}world. You must
strictly adhere to the requirements of the tasks I have provided to you.

### Environment
Provide a sequence of actions to navigate a world to reach a goal. (0,0) is located in the upper-
left corner and {(world_height,world_width)} lies in down-right corner.

### Rules
- <left = (0,-1)>
- <right = (0,+1)>
- <up = (-1,0)>
- <down = (+1,0)>

### Output Format
Actions = [action_0 action_1 … action_n]

Here are some examples:

###
Task: You are in a 10 by 10 world. There are obstacles that you must avoid at: (2,1). Go from
(0,1) to (3,4).

Actions = [right right right down down down]

###
Task: You are in a 10 by 10 world. There are obstacles that you must avoid at: (1,5) and (1,2).
Go from (5,4) to (0,5).

Actions = [up up up up up right]

###
Task: You are in a 10 by 10 world. There are obstacles that you must avoid at: (0,3), (2,5) and
(5,2). Go from (4,2) to (0,5)

Actions = [up up up right right up right]

### Task to Solve subtask_1

Task: {GTB_sub_task}

Actions =
```

