# OpenReview forum: "HSRL: Hierarchical Spatial Reasoning with Large Language Model"
_ICLR.cc/2026/Conference — ICLR 2026 Conference Withdrawn Submission_

### Official Review · Reviewer_pAbB · 2025-10-31

**Soundness:** 2
**Presentation:** 2
**Contribution:** 2
**Rating:** 4
**Confidence:** 3

**Summary:**

The authors highlight that existing Large Language Models (LLMs) excel in language understanding and abstract reasoning but consistently underperform in spatial reasoning. To address this issue, the paper proposes a hierarchical task decomposition approach for spatial problems along the dimensions of state and environment. This framework involves a higher-level LLM (Planner) that generates sequences of intermediate states (subgoals), while a lower-level LLM (Actor) defines local sub-environments and performs targeted planning for actions between each pair of subgoals. To mitigate the incompleteness of spatial reasoning in conventional LLMs and suboptimal partitioning problems, the authors introduce a training method that integrates Monte Carlo Tree Search (MCTS)-based exploration and experience with fine-grained advantage estimation via Group Relative Policy Optimization (GRPO).

**Strengths:**

Proposing a hierarchical task decomposition of spatial problems along the dimensions of state and environment to address the challenges LLMs face in spatial reasoning appears to be a novel contribution. Additionally, unlike most prior studies that utilize LLMs via APIs or in a frozen state, the decision to finetune them specifically for the target challenges stands out as a clear strength of this paper.

**Weaknesses:**

- The description of the proposed method could benefit from greater overall clarity. Although MCTS and GRPO form critical background components of the framework, the paper appears to lack references or explanations for them (e.g., in a preliminaries section). Furthermore, the linkage between the content in Section 3.2 and Algorithm 1 feels unclear; illustrating how outputs are structured at each step within the experimental setup would likely enhance the transparency of the authors' approach.
- While the finetuning strategy for LLMs differentiates this work from prior studies, it raises concerns about fairness in comparisons, as the HSRL's LLM receives information inaccessible to other baselines. Evidence from Table 2 shows that HSRL-Untrained results are similar to or lower than some baselines in Table 1, suggesting that HSRL's benefits may arise not from the hierarchical structure, but from an environment-optimized LLM informed by data unavailable to vanilla LLM baselines. Incorporating additional baselines or ablation studies to address these issues would be valuable.

**Questions:**

1. Line 259 states that “the LLM is prompted to generate subsequent potential states.” Does this expansion process involve repeatedly using the same prompt? If so, might it result in the LLM outputting only a single state? Similarly, for the “set of M candidate sequences” referenced in Line 302, I wonder if the same issue arises.
2. Line 317 refers to the “mean Q-value across all sibling states at depth n.” Does this assume that candidate paths have equal lengths? Is there any mechanism to regulate the lengths of paths generated by the LLM?
3. The paper mentions using the Qwen3-4B-Instruct-2507 model for HSRL. What motivated the selection of this specific model? I am curious whether performance varies with other LLMs.
4. Adding qualitative evaluations would strengthen the paper. For instance, visualizing the paths generated by HSRL during training in Figure 5 could more effectively illustrate the impact of M-GRPO. Likewise, examining step-by-step outputs in each environment as training advances would better showcase the paper's strengths.

---

### Official Review · Reviewer_LLiM · 2025-10-31

**Soundness:** 3
**Presentation:** 2
**Contribution:** 2
**Rating:** 6
**Confidence:** 3

**Summary:**

This paper introduces HSRL, a hierarchical spatial reasoning framework for LLMs that decomposes complex planning tasks at both the state and environmental levels. A key contribution is the M-GRPO algorithm, which integrates Monte Carlo Tree Search (MCTS) with GRPO-style policy optimization, using a novel fine-grained, node-level advantage function. The authors demonstrate state-of-the-art performance on several challenging benchmarks (Maze Navigation, Blocksworld, and GTB), showing significant improvements in completion rate and path optimality over a range of baseline methods.

**Strengths:**

1. The proposed HSRL framework effectively decomposes complex spatial planning tasks through a two-level hierarchy (state and environment), simplifying the problem structure.
2. The M-GRPO algorithm integrates MCTS exploration with GRPO policy optimization, employing a node-level advantage function for more precise credit assignment.
3. Experiments across three benchmarks (Maze Navigation, Blocksworld, GTB) demonstrates superior performance over exitsing prompting methods in completion rate and path optimality.

**Weaknesses:**

- The core idea of integrating tree-search with a fine-grained, node-level advantage function for GRPO has been previously explored in LLM reasoning, e.g., TreeRPO, TreeRL. The primary adaptation here appears to be the replacement of a tree-sampling method with MCTS, positioned as an improvement for spatial reasoning tasks.
- The baseline comparisons lack results from recent  powerful models like GPT-5/o3, Gemini 2.5, Qwen3 235B, or DeepSeek-R1.
- The presented baselines are largely untrained prompting methods (except for System-1.x). A critical missing ablation is a comparison against a non-hierarchical version of the GRPO-trained model. This is necessary to decouple the contribution of the GRPO fine-tuning (which is known to improve performance on verified reward tasks) from the contribution of the proposed hierarchical decomposition itself.

[1] TreeRPO: Tree Relative Policy Optimization, arXiv 2025.06
[2] TreeRL: LLM Reinforcement Learning with On-Policy Tree Search, arXiv 2025.06

**Questions:**

- What is the training time overhead of integrating MCTS into the GRPO loop compared to standard GRPO? Specifically, by what factor does the MCTS-guided exploration increase the time per training step/rollout?
- How does the search efficiency of MCTS degrade as the environment size increases? For instance, in a 30x30 maze, does the search time become prohibitive? Are there any strategies to maintain efficiency in significantly larger state spaces?
- How was the number of intermediate states (anchors) for the high-level planner determined? How the task completion rate and optimality are affected by varying this parameter?

---

### Official Review · Reviewer_yQ1a · 2025-10-31

**Soundness:** 3
**Presentation:** 2
**Contribution:** 3
**Rating:** 4
**Confidence:** 2

**Summary:**

This paper addresses spatial reasoning with large language models (LLMs) by introducing a novel algorithm MCTS-Guided Group Relative Policy Optimization (M-GRPO). The authors propose a hierarchical framework where a high-level policy, trained via M-GRPO, decomposes tasks into smaller subtasks solved by an untrained low-level LLM. The approach demonstrates improved performance over several baselines, including self-reflection and hierarchical reasoning methods, on spatial reasoning benchmarks.

**Strengths:**

1. The proposed hierarchical approach introduces a conceptually simple yet effective integration of MCTS-guided optimization with LLM reasoning.

2. The M-GRPO algorithm is both intuitive and simple.

3. The paper provides empirical improvements across several baselines, including self-reflective and hierarchical models.

4. Leveraging untrained low-level modules while optimizing high-level planning through reinforcement signals shows the strength of the proposed hierarchy.

**Weaknesses:**

1. Clarity could be improved. For instance, it is unclear whether high-level planning directly corresponds to the state decomposition process or represents a distinct layer of abstraction.

2. The training details are insufficient. The paper mentions a "complex reward function" for guiding M-GRPO training, but it is not evident whether similar reward formulations or fine-tuning procedures were applied to baselines. Such missing details make it hard to assess fairness and reproducibility.

3. The scope of evaluation appears inconsistent with the claims. "*these methods only consider high-level, coarse-grained task ...... such as robotic arm motion planning. Therefore, this study aims to fill this gap by solving the complex action planning problem*" While the paper criticizes prior work for focusing only on coarse-grained planning and states its goal as tackling fine-grained motion control (e.g., robotic arm planning), the actual experiments are limited to maze-like environments. This mismatch weakens the strength of the claims in the introduction and motivation.

**Minor Suggestions (Not affecting the score)**

1. Include a brief explanation of GRPO to make the paper more self-contained, especially for readers unfamiliar with the method.

2. Improve Figures 1 and 2, which currently occupy too much space at the cost of the manuscript clarity. Simplifying them or moving parts to the appendix could improve readability.

**Questions:**

1. Do the authors think that M-GRPO’s hierarchical mechanism is specific to spatial reasoning, or could it generalize to other domains and tasks?
2. In Table 2, what exactly does “HSRL (w/o MCTS)” denote? Does it refer to a variant fine-tuned only with GRPO? Additionally, what differentiates “State-Hierarchical Only” from “HSRL (untrained)”? A clearer explanation of the ablation design would help interpret the results.

---

> ### Author Response · Authors · 2025-11-27
>
> We thank the reviewer for the constructive feedback and for recognizing the novelty and effectiveness of our M-GRPO algorithm. We appreciate the opportunity to clarify the points raised regarding clarity, training details, and experimental scope.
>
> Response to Weaknesses:
>
> 1. Clarity of High-Level Planning: We clarify that our high-level planning directly corresponds to the state decomposition process, and it functions specifically to generate key sub-states. We apologize for the ambiguity and will revise the manuscript to explicitly state this relationship and improve the description of the framework.
>
> 2. Training Details and Fairness (Reward Function): Regarding the fairness of the comparison, we confirm that the ablation study "HSRL (w/o MCTS)" employed the exact same reward function as the M-GRPO method. The only difference was the removal of the MCTS guidance, using only standard GRPO for training. This ensures that the performance gap is attributable to the MCTS-guided exploration rather than differences in reward formulation. We will add these details to the experiment section to ensure reproducibility.
>
> 3. Scope of Evaluation (3D Environments): We acknowledge the reviewer's concern regarding the gap between our motivation (robotic arm planning) and the experiments (maze-like environments). The primary constraint is the lack of existing pure-text datasets suitable for describing 3D environments, which makes extending our method to 3D benchmarks challenging at this stage. However, we argue that the GTB environment is a multi-objective, complex game that presents significant challenges in action planning. We believe the results on GTB sufficiently demonstrate the effectiveness of our algorithm in solving complex planning problems. We will revise the manuscript to clarify that this work represents a preliminary exploration towards applying LLMs to 3D motion planning tasks, serving as a stepping stone for future research in more complex physical environments.
>
> Response to Questions:
>
> Q1: Generalization of M-GRPO: We believe the hierarchical mechanism of M-GRPO is generic and can be generalized to other domains. For tasks like mathematics, the high-level hierarchy (e.g., designing solution steps) is often clearer and more explicit than in spatial reasoning. We focused on spatial reasoning precisely because the hierarchical structure is less obvious and more challenging to extract compared to other domains. Therefore, success in this domain suggests strong potential for generalization to tasks with even clearer hierarchical structures.
>
> Q2: Clarification of Table 2 Definitions: We apologize for the confusion caused by the terminology in Table 2. We clarify the definitions as follows:
>
> HSRL (w/o MCTS): Refers to the variant trained without MCTS exploration, using only standard GRPO (Group Relative Policy Optimization) with the same reward function.
>
> State-Hierarchical Only: Refers to an untrained baseline that utilizes only the high-level state decomposition (High-level Hierarchy) without the environment decomposition step.
>
> HSRL (untrained): Refers to the full hierarchical model (including both State-level and Environment-level Hierarchies) but without any training.
>
> We will revise the captions and text in Table 2 to ensure these definitions are clearly communicated to the readers.

---

### Official Review · Reviewer_hgdL · 2025-11-01

**Soundness:** 2
**Presentation:** 2
**Contribution:** 1
**Rating:** 2
**Confidence:** 3

**Summary:**

This paper presents **HSRL**, a novel **hierarchical spatial reasoning framework** for large language models (LLMs). The key idea is a **two-level decomposition** of complex spatial planning tasks at both the *state* and *environment* levels, enabling LLMs to solve long-horizon problems through structured sub-tasking. To mitigate the sub-optimality of LLM-generated sub-goals, the paper proposes **M-GRPO**, a new post-training algorithm that integrates **Monte Carlo Tree Search (MCTS)** exploration with **fine-grained advantage computation** for node-level policy optimization. The method is evaluated on Maze Navigation.

**Strengths:**

1. The motivation for introducing a two-level hierarchy is sound. Spatial reasoning tasks often lack clear reward signals, and defining appropriate granularity for reasoning steps is challenging. By decomposing planning into state and environment levels, HSRL provides a structured formulation that helps reduce the search space and clarify the reward assignment problem. This design is well-motivated (Figure 2) and aligns with broader hierarchical RL principles.

2. Its combination of MCTS-based exploration (Eq. Page 5) and fine-grained advantage computation (Eqs. Pages 6–7) enables targeted improvement of planning policies. The pseudo-code in *Algorithm 1* is concise and reproducible.

3. Experiments cover diverse domains. Maze Navigation, Blocksworld, and GameTraversalBenchmark—evaluating not only completion and optimality rates but also **zero-shot transfer** capabilities (Table 1), underscoring the generalization potential of HSRL.

4. Providing complete prompt templates (Pages 15–22) improves reproducibility and clarifies implementation details for practitioners.

**Weaknesses:**

1. The integration of MCTS with LLM RL is a valuable research direction, but the key problem is how to design better tree search strategies to expand beyond UCT (through more advanced LLMs' evaluation or with model's perplexity itself), how to evaluate the reasoning process (works done on Process Reward, etc.), and eventually achieve advanced reasoning through LLM's self-evolution. This paper does not provide enough novel insights or techniques in this regard. Introducing MCTS into LLM reasoning is intuitive and not a new concept. This paper directly implements MCTS with the classical UCT strategy (Eq. Page 5) without significant modification or innovation.

2. I doubt the necessity of integration of GRPO in this framework. A strong baseline is lacking in the experiments: simply using MCTS to rollout and collect trajectories to SFT the LLM. And I found the M-GRPO is mathematically wrong. Directly applying the Q-value in MCTS to compute the advantage is problematic, since the sampled trajectories are limited, the evaluation of Q-value is inaccurate and is deemed to have high variance. The authors did not provide sufficient justification for this design choice, nor did they compare it with simpler alternatives.


3. Algorithmic Clarity:

- The definition of advantage computation ($A_{m,n}$, Page 6) ignores variability in trajectory lengths, which may bias learning.

- Algorithm 1 omits concrete termination criteria (IsSufficientlyDeep) and lacks specification of trajectory sampling and replay.

- The justification for omitting reward normalization (Page 7) is qualitative; quantitative evidence is needed to validate stability.

4. Domain Limitation and Real-World Transfer:
All experiments are in **grid-based or symbolic** environments. Despite claims of broader applicability, there are no tests in **3D or continuous domains** (e.g., embodied or robotic environments). Discussion of scalability and potential failure modes in these settings would strengthen the paper’s impact.

5. Generalization and OOD Claims:
While zero-shot transfer is evaluated, broader OOD claims (especially for high-dimensional or perceptual settings) seem **overstated** relative to current evidence.

**Questions:**

1. **Advantage Normalization:** What empirical evidence supports omitting reward normalization?
2. **Variable-Length Trajectories:** How is advantage computation stabilized for uneven trajectory depths?
3. **Real-World Extension:** Has HSRL been tested on 3D or perceptual tasks? If not, what are the main technical barriers?
4. **Trajectory Parsing and Failure Penalties:** What is the observed failure rate and its impact on learning dynamics?

---

### Note · Authors · 2026-01-05

**Comment:**

We have decided to withdraw our submission, "HSRL: Hierarchical Spatial Reasoning with Large Language Model", to ICLR 2026. After further internal review and consideration of the current manuscript's stage, we believe the paper requires additional experiments and refinements that cannot be completed within the current discussion period. We thank the program chairs and reviewers for their time and effort.

**Withdrawal Confirmation:**

I have read and agree with the venue's withdrawal policy on behalf of myself and my co-authors.